# Assessing the Reliability of SARS-CoV-2 Neutralization Studies That Use Post-Vaccination Sera

**DOI:** 10.3390/vaccines10060850

**Published:** 2022-05-26

**Authors:** Henning Jacobsen, Ioannis Sitaras, Marley Jurgensmeyer, Mick N. Mulders, David Goldblatt, Daniel R. Feikin, Naor Bar-Zeev, Melissa M. Higdon, Maria Deloria Knoll

**Affiliations:** 1Department of Viral Immunology, Helmholtz Centre for Infection Research, 38124 Braunschweig, Germany; 2W. Harry Feinstone Department of Molecular Microbiology and Immunology, Johns Hopkins Bloomberg School of Public Health, Baltimore, MD 21205, USA; 3International Vaccine Access Center, Department of International Health, Johns Hopkins Bloomberg School of Public Health, Baltimore, MD 21205, USA; mjurgen4@jhu.edu (M.J.); nbarzee1@jhu.edu (N.B.-Z.); mhigdon@jhu.edu (M.M.H.); mknoll2@jhu.edu (M.D.K.); 4Department of Immunizations, Vaccines and Biologicals, World Health Organization, 1211 Geneva, Switzerland; muldersm@who.int (M.N.M.); feikind@who.int (D.R.F.); 5Great Ormond Street Institute of Child Health, NIHR Biomedical Research Centre, University College London, London WC1E 6BT, UK; d.goldblatt@ucl.ac.uk

**Keywords:** COVID-19, vaccine, serology, antibody neutralization, SARS-CoV-2

## Abstract

Assessing COVID-19 vaccine effectiveness against emerging SARS-CoV-2 variants is crucial for determining future vaccination strategies and other public health strategies. When clinical effectiveness data are unavailable, a common method of assessing vaccine performance is to utilize neutralization assays using post-vaccination sera. Neutralization studies are typically performed across a wide array of settings, populations and vaccination strategies, and using different methodologies. For any comparison and meta-analysis to be meaningful, the design and methodology of the studies used must at minimum address aspects that confer a certain degree of reliability and comparability. We identified and characterized three important categories in which studies differ (cohort details, assay details and data reporting details) and that can affect the overall reliability and/or usefulness of neutralization assay results. We define reliability as a measure of methodological accuracy, proper study setting concerning subjects, samples and viruses, and reporting quality. Each category comprises a set of several relevant key parameters. To each parameter, we assigned a possible impact (ranging from low to high) on overall study reliability depending on its potential to influence the results. We then developed a reliability assessment tool that assesses the aggregate reliability of a study across all parameters. The reliability assessment tool provides explicit selection criteria for inclusion of comparable studies in meta-analyses of neutralization activity of SARS-CoV-2 variants in post-vaccination sera and can also both guide the design of future neutralization studies and serve as a checklist for including important details on key parameters in publications.

## 1. Introduction

SARS-CoV-2 has caused at least 500 million COVID-19 infections and 6 million deaths worldwide by May, 2022 [1]. Vaccines have been instrumental in controlling the spread of infection and drastically reducing disease severity and mortality [2]. The effectiveness of vaccination in preventing disease can depend on many factors, such as the specific vaccine being used, circulating variants, characteristics of the target population, etc., many of which are assessed by observational studies after the vaccines are approved for use [3,4]. However, obtaining results from clinical studies is a lengthy process, since a large enough number of cases for statistical significance needs time to accrue. For this reason, a quick assessment of vaccine performance against newly emerging SARS-CoV-2 variants is crucial. Neutralization assays are among the most valuable tools for evaluating viral immunity in vitro for many types of viruses including influenza, rabies, polioviruses, and MERS-CoV [5,6,7,8,9]. For this reason, these assays have been rapidly adopted for SARS-CoV-2, as they can provide early insights into potential reductions in vaccine performance [10,11,12]. Among the most useful such studies are in vitro neutralization studies that evaluate the ability of antibodies in post-vaccination specimens (e.g., sera) to bind and neutralize new SARS-CoV-2 variants, thereby preventing cellular infection [13,14]. Broadly speaking, there are four different types of in vitro neutralization assays: live virus neutralization; pseudo-virus neutralization; plaque-reduction neutralization; and focus reduction neutralization. A recent study evaluated all four types for SARS-CoV-2 and found each one to be robust, and the results comparable between assays [15].

However, such studies are not standardized in terms of design or methodology and thus pose a number of challenges when assessing them and trying to interpret their findings. For example, the reduction in neutralizing activity for a specific variant of concern following vaccination may range from 1-fold to 30-fold between studies [16], thus making the “true” degree of reduction unclear.

For over a year, we have been reviewing, evaluating and summarizing all available literature pertaining to SARS-CoV-2 variants and the respective neutralization activity of post-vaccination sera to inform WHO reports on vaccine performance in the context of emerging variants of concern [16]. Approximately 10% of the studies reviewed contained relevant neutralization data and were reported to the WHO, with pre-prints forming a large percentage of the earliest evidence. The lack of standardization of study design, methods and reporting quality posed a particular challenge in the interpretation of the results. We experienced increased uncertainty when attempting to compare studies with missing or poorly described study design or reporting parameters. For example, some articles did not report subject characteristics (such as age, sex, prior infection status) or did not stratify the results by important sub-groups (e.g., patients with clinical conditions affecting immune responses versus otherwise healthy patients, or young patients versus elderly patients). In other cases, the methods used were not always fully described, such as the quantity of virus used in neutralization assays, how pseudo-viruses were constructed and whether they contained the full complement of Spike protein mutations. In the race to contribute knowledge that could assist in arresting the pandemic, it is understandable that many of these details may have been overlooked. However, this inevitably renders a substantial number of otherwise valuable studies uninterpretable. These details are especially important for understanding the reasons for heterogeneous results between studies and for identifying which results are the least biased.

We expect that data coming from studies with high reliability will be less biased and thus more informative. To assist our work, we developed a tool that assesses the overall reliability of studies reporting neutralization activity of post-vaccination sera against SARS-CoV-2, where reliability was defined as a measure of methodological accuracy, proper study setting concerning subjects, samples and viruses, and reporting quality. The assessment is based on the presence or absence of key study design parameters and information pertinent to interpreting the results. In this manuscript, we describe the key parameters that neutralization studies should address, including the rationale for each and the impact they may have on study reliability. Finally, we present a tool to apply these parameters in a standardized way and that summarizes the overall level of reliability of the study across parameters.

## 2. Methods

We first aimed to identify and characterize aspects that might affect the reliability of results in studies assessing neutralizing antibody titers against SARS-CoV-2 in post-vaccination sera. Aspects identified included those with potential to affect the study outcome either because of technical limitations, such as poor assay standardization, statistical limits, such as small sample number, or because of insufficient or absent reporting, which prevents proper interpretation and evaluation of the reliability of results. Each aspect identified was described using specific, targeted parameters. All aspects and parameters were identified using our combined expertise in the design, conduct and analysis of neutralization assays, most importantly, through our experience screening and synthesizing all SARS-CoV-2 neutralization literature available since March 2021, and through extensive discussions and collaboration with recognized experts in this field [16].

### Development of the Reliability Assessment Tool (RAT)

For a more convenient application of our method, we created an excel-based reliability assessment tool (RAT). The RAT required responses for each of 33 parameters; the nature of the response determines the reliability level linked to that particular parameter. We defined possible responses for each parameter, such as “not reported”, “yes” or “no”. For each parameter response, we considered whether its impact was likely to be low, medium or high on neutralization study reliability. For parameters with no information reported, an assessment of reliability of these studies is not possible, thus their reliability was defined as unclear. For example, the sampling period parameter assesses if all samples were collected at least 7 days post full vaccination (second vaccination in the case of two-dose regimens), and responses “yes”, “no” or “not reported” were assigned an impact on reliability of “no”, “high” or “unclear”, respectively. Some parameters with stratifications can have more complex outcomes. For parameters not applicable to all studies, “not applicable” is included as a response option and is assigned “no” impact on reliability when relevant.

The overall impact on reliability of each aspect is determined by the lowest reliability identified in any of that aspect’s parameters, with the order “no” < “low” < “medium” < “unclear” < “high” impact on reliability. Taking the “clinical characterization” aspect as an example, if the parameter “relevant clinical characterization reported” is reported or is not applicable, the result is “no impact on reliability”. However, if the results are not stratified for immunocompromised individuals (additional parameter for this aspect), the result is a “high impact on reliability”. Consequently, the overall aspect impact on reliability will be equally high. In a similar way, the overall reliability of a study is calculated as the lowest reliability recorded among 11 aspects. The association between parameter responses and reliability gradation allows for consistent and objective evaluation of each study and fair comparisons across studies.

To illustrate the application of the RAT, we selected 10 publications that we had previously screened for neutralization antibody titer results in the context of SARS-CoV-2 variants [16]. For reasons of consistency, we selected studies that provided data on Pfizer BioNTech-Comirnaty-vaccinated individuals against SARS-CoV-2 variant B.1.361 (Beta) and that included fold-changes of neutralization against this variant (compared to the parental strain) ranging from the highest changes reported to the lowest. The data for the assessment of each study are provided in Appendix A. We applied the RAT to all 10 studies and assessed the reliability of each aspect, as well as the overall study reliability.

## 3. Results

We identified three categories of study characteristics as important in ensuring the reliability of neutralization assay results: cohort details, assay details and data reporting details. Each comprised multiple aspects with potential to affect the results, including sample size, previous SARS-CoV-2 infections, vaccination regimen, sample collection period, detailed demographic characterization (while maintaining strict anonymity and protecting identifiable data), clinical history of the investigated subjects (with a special focus on pre-vaccination COVID-19), important technical details about the viruses and samples used, and how neutralization assays were performed and how details about the data were reported. A brief description of the 33 parameters identified for the various aspects and the rationale for each is shown in Table 1. Some parameters identified are relevant to many types of studies (e.g., sample size, age distribution, unbiased cohort selection, etc.), but some are specific to neutralization assays using post-vaccination sera. More detail including information on how the reliability assessment per parameter was assigned is provided in Appendix A. The RAT that facilitates assessing the study’s reliability is available for download for direct and free use (Appendix A). A graphic overview on how to use the tool is provided in Appendix A.

The results of our 10-study reliability assessment using the RAT are shown in Figure 1. One of the ten studies evaluated was assessed as ‘unreliable’. In this case, assay standardization was found to be insufficient because the authors accepted a variance of 500% in the virus input for neutralization assays. This study reported a much higher fold reduction in neutralization compared to all the other studies, suggesting this outlier response may be due to the insufficient assay standardization. Three of the ten studies lacked essential information in one or more aspects; these were the aspects of assay standardization (virus input), demographic characterization of the study cohort (age distribution) and clinical characterization (health status of subjects). The remaining six studies were deemed to have low or medium impact on reliability.

## 4. Discussion

To facilitate the review and interpretation of time-sensitive information on vaccine effectiveness against emerging SARS-CoV-2 variants using post-vaccination neutralization antibody titers, we identified and characterized aspects that may affect the reliability of these studies. Identifying studies that may have moderate or high risk of bias is important when considering their results in the context of the pool of studies and might provide insights into some of the causes of heterogeneity when observed. Many neutralization studies produce important data that might help tackle the current pandemic, and we expect most are conducted with the highest technical and analytical standards. Yet, we often observe that technical methodological details or critical information on the study cohort are absent from the manuscript, especially in short communications. This creates a missed opportunity because it automatically diminishes the reliability of the reported results when the reader cannot informatively assess the study.

The list of parameters could serve as a guidance checklist for future publications on the type and level of detail needed to ensure their study’s contribution to the literature is maximized, as well as inform on study design and technical methodologies. Studies assigned high reliability by the RAT may be more likely to be included in meta-analyses or have their results being taken into consideration in policy-making decisions. We therefore strongly encourage authors to consider the list of key parameters as publication guidelines and to address all applicable aspects.

We designed an easy-to-use tool (the RAT) that assesses the overall reliability of these studies to facilitate synthesis and meta-analysis of study results by identifying studies with potentially high or unclear impact on reliability to consider their exclusion or down-weighting. We developed the RAT to allow an objective, structured and comparable assessment of the reliability of neutralization studies. Applying the RAT to ten relevant example studies demonstrated that it is able to distinguish studies on the basis of degrees of completeness and reliability of key parameters and identify which parameters within each study may lower its reliability. The RAT could be used to screen a wider body of available literature on SARS-CoV-2 post-vaccination neutralization to pinpoint the most common aspects responsible for reducing reliability or comparability. Specific recommendations could follow to avoid common mistakes and to harmonize studies across the globe.

This tool has several limitations. First, no experimental validation was performed to confirm the impact of these aspects. However, the aspects and their possible impact on reliability were identified and assessed based on our scientific expertise, our particular experience in screening all pertinent literature in the past year, as well as discussions with other experts in the field. Second, studies assessing post-vaccination neutralization antibody titers against SARS-CoV-2 cover a broad spectrum of clinical, technical or epidemiological settings, and not all aspects that are used in this tool may always apply to the specific setup of a study. Therefore, it is important to consider the study setting when using this tool. To address this limitation, we encourage a differentiated reporting which will identify the strengths and weaknesses of a particular study not only by considering the overall study reliability assessment, but also by focusing on the reliability of each individual aspect. This allows highlighting certain categories that may be more applicable or important than others in the study-specific setting. It is important to mention that, while we strived to develop the tool in a way that it is easy to use, it requires the assessment of technical details such as cell culture-associated effects or details about the neutralization assay itself, which may require the user to have at least a basic understanding of virological and serology techniques.

## 5. Conclusions

We hope that this tool will contribute to the usefulness of SARS-CoV-2 neutralization literature by identifying studies with low impact on reliability to target for evidence synthesis and to guide future studies in improving the impartiality of the results they report.

## Figures and Tables

**Figure 1 vaccines-10-00850-f001:**
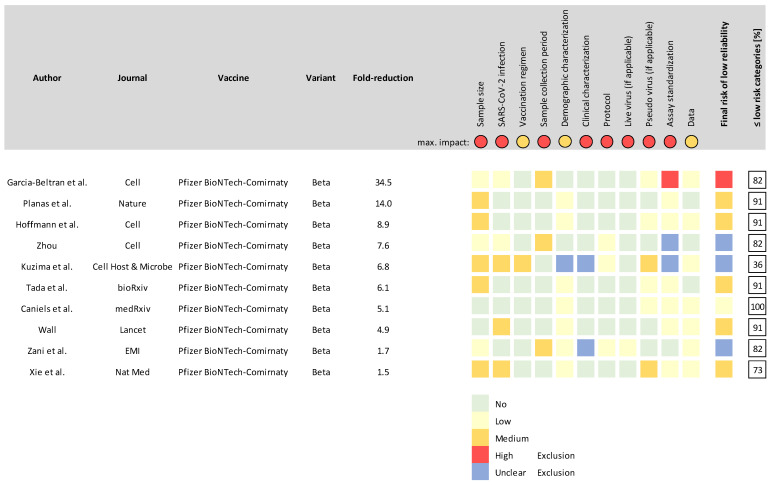
Reliability assessment of ten selected studies reporting post-vaccination neutralization antibody titers against SARS-CoV-2 as determined using the RAT. Ten studies reporting fold-changes in neutralization capacity against the Beta variant and reflecting the full spectrum of the reported fold changes were assessed with the RAT. The aspect-specific impact on reliability is indicated in color-coded boxes on the right of the study, and the overall risk of low reliability is indicated in the same way as “Risk of low reliability”. In the last column, the cumulative percentage of aspects that returned only “no” or “low” risk of low reliability for each study is reported to further stratify the overall reliability.

**Table 1 vaccines-10-00850-t001:** Aspects and parameters that are associated with possible risks of low reliability for studies assessing post-vaccination neutralization against SARS-CoV-2.

Cohort Details	
Sample Size	
Sample size	Required to assess the statistical strength, potential for spurious results and overall generalizability of the results. Reduces the probability of spurious results.
SARS-CoV-2 Infection	
Reported	There is accumulating evidence that convalescent subjects develop a stronger immune response to vaccination compared to SARS-CoV-2-naïve subjects [17,18,19].
Confirmed	Because of the potential impact of non-naïve subjects, the cohort should be screened for previous COVID-19 by highly sensitive methods (e.g., NP-ELISA or by repeated qPCR screening over the whole study period and pre-study period if applicable).
Breakthrough cases reported	Especially in longitudinal studies, breakthrough cases of COVID-19 might occur. These infections can affect the subject’s immune response and the neutralization titers because of boosting-like effects.
Breakthrough cases stratified	If breakthrough cases of COVID-19 are reported for the study cohort, the neutralization results should be stratified for naïve and infected subjects to acknowledge booster effects of the infection.
Vaccination Regimen	
Dosing interval reported	There is increasing evidence that the dosing interval for vaccines with a prime-boost regimen can affect the immune response, including neutralization titers [20,21].
Stratified by partial/full immunization	Certain studies investigate neutralization titers from partially and fully vaccinated individuals. It is imperative that these cohorts are completely separated, as it is known that titers from partially immunized subjects are significantly inferior to titers from fully immunized subjects [22,23,24].
Sample Collection Period	
≥7 days post last dose	Because of the kinetics of neutralizing antibody generation, no samples taken ≤7 days post immunization should be considered [25,26,27].
Stratified OR ≥14 days and ≤4 months post last dose	Peak neutralization titers are usually observed 14 days post immunization followed by a gradual decline of neutralization activity (waning) [25,26,27]. When assessing neutralization results and especially when comparing studies, it is important to acknowledge these kinetics by stratification of the results or by only including subjects sampled within a range of peak titers.
Demographic Characterization	
Age distribution reported	As for many other pathogens, age is very likely to also affect neutralization titers against SARS-CoV-2, especially when imperfect responses are reported [28,29,30].
Stratified by age group	To acknowledge the possible effects of age on neutralization titers, we recommend stratifying the results based on age groups, especially for older adults (≥60 years), adults and children (<18 years).
Sex distribution reported	Although there are conflicting data, several studies suggest that the biological sex might also affect the neutralization titers against SARS-CoV-2 [31,32].
Stratified by sex OR equal sex distribution	To acknowledge possible effects of the biological sex on neutralization titers, we recommend stratifying the results based on the subjects’ sex.
Cohort selection unbiased	If neutralization titers are generally assessed, it is essential that no biased pre-selection (for example, high responders only) was performed on the study cohort.
Study period and geographic location reported	To correctly interpret SARS-CoV-2 infections occurring before or during the study, it is important to understand which SARS-CoV-2 variants caused infection, because variants can have differential effects on the neutralization response [33]. If the variant distribution is not available, the study period and geographic location allow predicting a likely distribution of the variants.
Variant prevalence reported	As described above, the prevalence of variants can help to understand and to correctly interpret data in the context of SARS-CoV-2 infections that occurred during or before the study period.
Stratified by variant prevalence	We recommend stratifying the results by the respective variants causing infection to acknowledge emerging data on potential effects of SARS-CoV-2 infection on cross-neutralization response in vaccinees [33].
Clinical Characterization	
Reported	Many study subjects are likely to have clinical characteristics that might affect the post-vaccination immune response, such as immuno-suppression (more likely in older adults), frailty (more likely in women) or pregnancy (women of reproductive age only). Relevant clinical characteristics of the study cohort must be reported.
Stratified by immuno-compromised	If a clinical characterization is reported, we highly recommend stratifying the results for immuno-compromised subjects, as they might significantly affect the overall neutralization titers in a cohort [34,35].
Assay Details	
Protocol	
Assay type reported	It is imperative to provide the assay type (live virus neutralization, pseudovirus neutralization, plaque-reduction neutralization, etc.) along with the determined endpoint (NT20, NT50, NT80 etc.), as both can affect the neutralization titer [15,36,37].
Precise protocol reported	A precise assay protocol can help to correctly interpret the results and to understand possible differences among studies.
Live Virus Strain (if Applicable)	
Virus lineage reported	If a live virus is used for neutralization, the lineage and origin must be reported to allow a correct interpretation of the results.
Confirmation by sequencing	SARS-CoV-2 can acquire adaptational mutations in cell culture passaging [38,39,40,41]. Because it is not yet known if these mutations might affect neutralization titers, the virus sequence should be confirmed for the passage used in neutralization assays.
Pseudo Virus Strain (if Applicable)	
Construct details reported	If a pseudovirus is used for neutralization, details on pseudovirus construction and origin must be reported to allow a correct interpretation of the results.
All variant-associated spike mutations	To properly assess antibody neutralization against SARS-CoV-2 variants using a pseudovirus system, it is important that the virus construct contains at least all spike mutations that are associated with the respective variant. We recommend https://covdb.stanford.edu/as (accessed on 10 April 2022) a reference.
Confirmation by sequencing	To follow good scientific practice and to provide maximum credibility of the assay, we recommend confirming the pseudovirus sequence (not the plasmids) by sequencing prior to use in neutralization assays.
Assay Standardization	
Virus titer reported and consistent	With a neutralization assay, the capability of the subjects to neutralize a defined amount of virus is measured. Standardization of input virus is essential to provide high-quality results. The variance accepted for the virus input translates into the variance of the neutralization titer and determines the sensitivity and resolution of the assay.
Error in titer reported by back titration	The virus input for each assay performed can be easily assessed by back titration. This allows a precise description of the variance conferred by the virus input and therefore an optimal assessment of the assay results.
WHO international standard antibody used	By now, the WHO international standard antibody is available to allow the standardization of the neutralization results for SARS-CoV-2 neutralizing antibodies [42]. This standardization can enhance the comparability of results and can support an optimal interpretation of the results.
Details on cell culture reported	Neutralization assays are performed in a cell culture; the virus infectivity is highly dependent on the target cells and can be influenced by many factors such as cell confluency, passage number, contamination, temperature and many more. We therefore recommend reporting cell culture techniques as detailed as possible.
Data	
Data Reporting	
Raw data reported	Direct reporting of raw data (ideally linked to the respective subject information such as age, sex, etc.) supports an optimal interpretation of the results. Furthermore, raw data can be used to confirm or re-analyze statistics, if applicable.
Reference virus is appropriate	In some studies, fold changes are calculated. For this, it is important that comparisons are always made using the vaccine seed strain as a reference, since the homologous comparison will determine the baseline neutralization activity of the sera and any antigenic differences between the vaccine strain and other variants [43].
Data shown as individual data points with statistics	Appropriate presentation of data and statistics can support correct interpretation of the results and re-analysis as applicable. The sole presentation of, for example, fold changes or bar graphs without presentation of data distribution adds uncertainty to the results and does not allow for optimal assessment.

## Data Availability

Not applicable.

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
