# Peer review of "Assessing the Reliability of SARS-CoV-2 Neutralization Studies That Use Post-Vaccination Sera"

_vaccines, 2022, doi:10.3390/vaccines10060850_

Round 1

Reviewer 1 Report

The manuscript submitted by Jacobsen et al. entitled "Assessing the Reliability of SARS-CoV-2 Neutralization Studies Using Post-Vaccination Sera" gives novel insights to provide selection criteria for the inclusion of comparable studies in meta-analyses related COVID-19 studies that aim to analyze the vaccine effectiveness against emerging SARS-CoV-2 variants. Furthermore, this manuscript also gives tools to guide the design of future neutralization studies, which is a very important goal, regarding the current diffuse situation of the studies that evaluate the immune responses after vaccination. In sum, this manuscript once published, it will have great utility to the scientific community. 

Author Response

We thank the reviewer for the positive and encouraging feedback. No response required.

Reviewer 2 Report

The manuscript by Henning Jacobsen and colleagues employed the evaluation strategy of reliability for SARS-CoV-2 Neutralization studies. The author set eleven criteria to assess the reliability of clinical study data detected SARS-CoV-2 neutralization titer. The evaluation strategy was detail and expected to expand the applicable scopes. The author proposed that the tool provides explicit selection criteria for inclusion of comparable studies in meta-analyses, and can also guide design of future neutralization studies. The study is a novel attempt, but a lot of questions exist that must be answered to improve the meaning of this study.

Major Comments:

  • Study power—How did the author determine that the three aspect categories (cohort details, assay details, and data) were the main factors that could bias the reliability of Serum neutralizing antibody detection. This seems quite groundless and should take into account the purpose of the study and the number of subjects enrolled.
  • Method—The assessment criteria were based on the expertise of the author, that might be one-sided and limits the meaning of the study. The differences among the reliability evaluation tools involves this study, Cochrane and PRISMA should be included in the paper.
  • Reliability Assessment Tool (RAT) the definition might be inappropriate, should be defined as Reliability Assessment Lists for the lack of Algorithmic stuff. The title and the full text should be rearranged around the “Lists”.
  • The ten studies assessing the reliability were just a demo for checklist evaluation, but failed to demonstrated the accuracy and sensitivity of evaluation checklist. Another evaluation strategy or algorithm should be applied to make a comparison identifying the priority of the reliability evaluation criteria.
  • Grammar – The manuscript needs English revision.

Author Response

Major Comments:

Study power—How did the author determine that the three aspect categories (cohort details, assay details, and data) were the main factors that could bias the reliability of Serum neutralizing antibody detection. This seems quite groundless and should take into account the purpose of the study and the number of subjects enrolled.

We thank the reviewer for highlighting that this important aspect was not sufficiently explained. We have now addressed this question more comprehensively in the manuscript (lines 98 – 102). Briefly, to determine the aspects and parameters that comprise the RAT, we relied on a) published literature (see Table 1 in which references are provided to justify why many of the parameters were included), b) our combined (multiple authors) expertise and experience in both conducting these types of studies and in meta-analyses across studies, and c) our observations from our extensive literature review, which revealed patterns in under-reported study design parameters that were relevant for our meta-analyses. The under-reported parameters are those that can influence results and therefore create heterogeneity across studies and add uncertainty in meta-analytic estimates.

We agree that the parameters of interest depend on the research questions addressed. The RAT was developed with assessment of studies evaluating post-vaccination sera in neutralising SARS-CoV-2 variants in mind only. We have now clarified this throughout the manuscript. The assessment applies to all studies presenting such data, whether primary or secondary objective.

Method—The assessment criteria were based on the expertise of the author, that might be one-sided and limits the meaning of the study. The differences among the reliability evaluation tools involves this study, Cochrane and PRISMA should be included in the paper.

The choice of assessment criteria has been addressed above and has now been included in the manuscript in a more comprehensive manner. Cochrane and Prisma are important tools; however, they are used for systematic literature reviews and meta-analyses, which are not the goals of our study. In contrast, our manuscript identifies a list of parameters that must be considered when designing, reporting or evaluating individual neutralization experiments.

Reliability Assessment Tool (RAT) the definition might be inappropriate, should be defined as Reliability Assessment Lists for the lack of Algorithmic stuff. The title and the full text should be rearranged around the “Lists”.

We would kindly refer the reviewer to the actual RAT tool provided as supplementary data 3. We believe that the term “tool” is appropriate, since a) each parameter is assigned a different weight in its effect on the reliability of the results (calculated automatically according to the answer the user gives), b) the RAT allows for a uniform and unbiased assessment of a study’s reliability (because it contains a set range of answers), c) the RAT is interactive, and d) the RAT will become publicly-available with the publishing of the manuscript.

The ten studies assessing the reliability were just a demo for checklist evaluation, but failed to demonstrated the accuracy and sensitivity of evaluation checklist. Another evaluation strategy or algorithm should be applied to make a comparison identifying the priority of the reliability evaluation criteria.

We apologize for the misunderstanding in what the ten studies (Figure 1) represent and we have tried to make it more clear in the manuscript. Briefly, the ten studies included in Figure 1 are intended to illustrate how the RAT can identify when parameters important in considering the results are under-reported or are not used properly (as in one of the studies where a 500% variance in virus input was used) and how real studies vary in the nature and extent of reporting on these parameters. It is not intended to be representative of the status all studies, which we plan to evaluate in the future.

We also added an additional paragraph discussing the results from figure 1 to clarify our rationale (please refer to new “results” section) and where we show that the tool possesses sensitivity. It is currently not possible to assess accuracy of our tool because of a missing comparator because our tool is novel. However, we agree that this will be important in the future.

Grammar – The manuscript needs English revision.

We have taken this into consideration, and we corrected any mistakes in grammar or syntax we have identified.

Reviewer 3 Report

Estimated Authors of the paper "Assessing the Reliability of SARS-CoV-2 Neutralization Studies Using Post-Vaccination Sera",
I've read your article with great interest. In the present paper, a comparison between neutralization studies should have been provided, then guaranteeing the development of a new screening tool that may be of some interest for scientific researchers and public health stakeholders in the future events of the ongoing pandemic. 
As succinctly but clearly represented in the introduction section, neutralization studies have not benefited from a standardized design, therefore a comparison between various studies is particularly difficult.
As a consequence, the results of this study and more precisely the RAT item provided by this study may be quite useful.
Unfortunately, the status of this paper if far from being optimal, and consistent with the quality requirements of Vaccines, for the following reasons:
1) no information is provided neither on the data source nor on the results; Authors vaguely describe the rationale behind this study (rows 63 to 77), but no actual and accurate "methods"section is provided, impairing a future replication of this study, particularly when dealing with the source for retrieved data.
2) items included in RAT are not described across the text, requiring the reader to rely on supplementary material
3) no actual data reporting is provided, and no description of the results reported in Figure 1 is provided
4) discussion is therefore very vague and general, and does not provide any information on the actual results of this research

In other words, I recommend the Editors of Vaccine to reject this paper in its current version. Authors could improve the present study by following the subsequent recommendations:
1) provide an accurate "materials and methods" section that must provide the following information: how the data were retrieved from international literature (i.e. was a systematic review performed? Reference 15 does not lead to a specific article nor to a grey literature article: I've found the following document, https://view-hub.org/sites/default/files/2021-11/Neutralization%20Plots_1.pdf: please provide both a full link and a description, even if summary, on the research strategy);
2) describe in full details the results that are provided in Figure 1.
3) discuss your results accordingly and specifically on the basis of the actual results of your study, then generalize your research discussing the implication of your new instrument for public health research

Author Response

Estimated Authors of the paper "Assessing the Reliability of SARS-CoV-2 Neutralization Studies Using Post-Vaccination Sera",

I've read your article with great interest. In the present paper, a comparison between neutralization studies should have been provided, then guaranteeing the development of a new screening tool that may be of some interest for scientific researchers and public health stakeholders in the future events of the ongoing pandemic.

It was unclear to us what was meant by “comparison between neutralization studies” that should have been provided. We added a comparison between four different types of in vitro neutralization assays to the manuscript (lines 50 – 54).  But perhaps this refers to a comparison of the results between different studies to show the heterogeneity and how study design might explain that.  If so, there is already text explaining that (please refer to introduction). Or perhaps it meant that the results of the RAT applied to several studies should be compared; if so, that is addressed by Figure 1 for a sample of studies and to do for all studies is beyond the scope of this manuscript, but is planned for the next steps.

As succinctly but clearly represented in the introduction section, neutralization studies have not benefited from a standardized design, therefore a comparison between various studies is particularly difficult.

As a consequence, the results of this study and more precisely the RAT item provided by this study may be quite useful.

Unfortunately, the status of this paper if far from being optimal, and consistent with the quality requirements of Vaccines, for the following reasons:

1) no information is provided neither on the data source nor on the results; Authors vaguely describe the rationale behind this study (rows 63 to 77), but no actual and accurate "methods"section is provided, impairing a future replication of this study, particularly when dealing with the source for retrieved data.

We apologize for any confusion. This is not a study but rather a proposed approach to evaluate a set of key parameters that are likely to influence results of neutralization studies.  The data sources that helped us identify the parameters are included as references in Table 1, and Supplementary Table 1. The references for the 10 example studies we used to illustrate the application of the RAT are included in Supplementary Data 2, but we have now included them in the main manuscript also. We also made finding the methods section of the manuscript more clear by adding a header ‘Methods” to the start of that section.

2) items included in RAT are not described across the text, requiring the reader to rely on supplementary material

Thank you for this comment. We discussed this question among the authors and feel that a full discussion on all items important for the RAT in the main text would disrupt the manuscript’s structure. Hence, we decided to include a brief description of all items in the main text (Table 1) and put the comprehensive overview in the supplementary material. We also thought that the organization of a table format would best present each item in this long list since content can get buried when in paragraph form. We were also constrained by the Journals’ word limit, which would be exceeded by moving the Supplemental Materials content to text.

3) no actual data reporting is provided, and no description of the results reported in Figure 1 is provided

The only data acquired within this work are the evaluation results of the ten example studies assessed by the RAT tool (figure 1). These data are reported in supplementary data 2 in form of the checked RAT tool for each study. We refer to this in lines 135-136.

We added an indicator (“Figure 1”) to text when referring to that figure, and added additional  description of the results (please refer to the new “results” section).

4) discussion is therefore very vague and general, and does not provide any information on the actual results of this research

Thank you for this feedback.  We have revised the discussion accordingly throughout

In other words, I recommend the Editors of Vaccine to reject this paper in its current version. Authors could improve the present study by following the subsequent recommendations:

1) provide an accurate "materials and methods" section that must provide the following information: how the data were retrieved from international literature (i.e. was a systematic review performed? Reference 15 does not lead to a specific article nor to a grey literature article: I've found the following document, https://view-hub.org/sites/default/files/2021-11/Neutralization%20Plots_1.pdf: please provide both a full link and a description, even if summary, on the research strategy);

Thank you for your recommendations. We have added a ‘Methods’ header to identify that section.  We would like to highlight that this work is not a systematic literature review or meta-analysis, but aims to provide a much-needed and scientifically-relevant basis for the evaluation of neutralization studies. The items that we identified as important for the reliability of studies on neutralization data were identified via an ongoing classical literature review for another purpose, but the lit review itself was not the purpose of this manuscript. 

The parameters identified included some previously described as being important (references included in Table 1) and the remaining were based on the experience and expertise of multiple authors as part of the lit review for a meta-analysis on neutralization studies (separate body of work than what is presented here). This strategy is reported in lines 91 – 102, which we expanded based on your suggestions and separated it from the main text as a “methods” section. We also fixed the indicated reference (now Reference No. 16) – thank you for identifying the error.

2) describe in full details the results that are provided in Figure 1.

Please see above.

3) discuss your results accordingly and specifically on the basis of the actual results of your study, then generalize your research discussing the implication of your new instrument for public health research

We have edited the results section to provide additional clarity on the purpose and findings of this work to identify relevant reliability parameters and develop a tool for others to use to assess studies. We have also revised the manuscript throughout to clarify the implications of this work.

Reviewer 4 Report

The manuscript by Jacobsen, Sitaras et al. is a meta-style analysis reporting a tool/methodology for standardized analysis and comparison of SARS-CoV-2 vaccine efficacy through viral microneutralization assays. The authors highlight three critical components required for reliability determination in such studies, including rigorous detailing/reporting on: Cohort Details/Size, Details of the in vitro assay conducted, and availability of the data in both raw and processed forms.

The proposals highlight by the authors seem sensible and complete and, as highlighted by an example comparison of several contemporary studies analyzing patient serum neutralization of SARS-CoV-2 beta variant, immediately highlight relative strengths and weaknesses between studies as well as allowing quick assessment of the reliability of the reported data. Such tools/methods are vital in the ongoing effort against the COVID-19 pandemic, particularly in the face further potential variants of concern.

Minor Points:

While it is unlikely since most published studies are well anonymized, the authors’ attention to detail, particularly around the area of cohort demographic data, must also recognize the need to protect/limit potentially identifiable data, and a statement to recognize this would be well advised.

Page 4, table, 4th subheading: ‘life virus…’ should read “live virus…”

Author Response

The manuscript by Jacobsen, Sitaras et al. is a meta-style analysis reporting a tool/methodology for standardized analysis and comparison of SARS-CoV-2 vaccine efficacy through viral microneutralization assays. The authors highlight three critical components required for reliability determination in such studies, including rigorous detailing/reporting on: Cohort Details/Size, Details of the in vitro assay conducted, and availability of the data in both raw and processed forms.

The proposals highlight by the authors seem sensible and complete and, as highlighted by an example comparison of several contemporary studies analyzing patient serum neutralization of SARS-CoV-2 beta variant, immediately highlight relative strengths and weaknesses between studies as well as allowing quick assessment of the reliability of the reported data. Such tools/methods are vital in the ongoing effort against the COVID-19 pandemic, particularly in the face further potential variants of concern.

Minor Points:

While it is unlikely since most published studies are well anonymized, the authors’ attention to detail, particularly around the area of cohort demographic data, must also recognize the need to protect/limit potentially identifiable data, and a statement to recognize this would be well advised.

We would like to thank the reviewer for making this important point. We have now made a statement to this effect (lines 145 - 146).

Page 4, table, 4th subheading: ‘life virus…’ should read “live virus…”

We have corrected this. Thank you.

Reviewer 5 Report

Jacobsen and colleagues wrote an important paper, well structured and well presented. Below my minor suggestions

  1. Introduction: updata data on SARS CoV2 at the day of resubmission.
  2. Please write a methods section
  3. Figure and tables are clear and increase the quality of paper
  4. Discussion : add future perspectives that come from your interesting paper.
  5. Explain better how confirm the performance of RAT
  6. Minor langauge mistakes are present. Please revised it

Author Response

Jacobsen and colleagues wrote an important paper, well structured and well presented. Below my minor suggestions

Introduction: updata data on SARS CoV2 at the day of resubmission.

Thank you for your positive and constructive feedback. We have updated the respective COVID-19 data.

Please write a methods section

Thank you for your suggestion. The methods section now stands out from the rest of the manuscript.

Figure and tables are clear and increase the quality of paper

Discussion : add future perspectives that come from your interesting paper.

The Discussion section includes future perspectives, which have now been expanded based on your suggestion.

Explain better how confirm the performance of RAT

Because the RAT is the first tool available to systematically assess the reliability of studies on neutralization responses and no comparative approaches are yet available, it is currently unfeasible to assess performance of our tool. In figure 1 we show that our tool allows a differential and sensitive assessment of parameters affecting study reliability and we discuss the implications of this. This however is an important question to address in the future when alternative or complementary tools are available.

Minor langauge mistakes are present. Please revised it

We have attempted to correct the mistakes in grammar or syntax we have identified.

Round 2

Reviewer 3 Report

Estimated Authors,

thank you for having accurately and precisely replied to all my previous comments, even to those that may have been perceived as difficult to understand (see point 1).

In fact, all my concerns and doubts have been addressed, and the improvements you've performed have solved all remaining doubts about the aims and design of this paper, whose acceptance I'm therefore endorsing.